

# *Porphyrobacter mercurialis* sp. nov., isolated from a stadium seat and emended description of the genus *Porphyrobacter*

David A. Coil[1], Jennifer C. Flanagan[1], Andrew Stump[1], Alexandra Alexiev[1], Jenna M. Lang[1] and Jonathan A. Eisen[1,2,3]

[1] Genome Center, University of California, Davis, CA, United States
[2] Department of Ecology and Evolution, University of California, Davis, CA, United States
[3] Department of Medical Microbiology and Immunology, University of California, Davis, CA, United States

## ABSTRACT

A novel, Gram-negative, non-spore-forming, pleomorphic yellow-orange bacterial strain was isolated from a stadium seat. Strain Coronado[T] falls within the *Erythrobacteraceae* family and the genus *Porphyrobacter* based on 16S rRNA phylogenetic analysis. This strain has Q-10 as the predominant respiratory lipoquinone, as do other members of the family. The fatty acid profile of this strain is similar to other *Porphyrobacter*, however Coronado[T] contains predominately C18:1$\omega$7cis and C16:0, a high percentage of the latter not being observed in any other *Erythrobacteraceae*. This strain is catalase-positive and oxidase-negative, can grow from 4 to 28 °C, at NaCl concentrations 0.1–1.5%, and at pH 6.0–8.0. On the basis of phenotypic and phylogenetic data presented in this study, strain Coronado[T] represents a novel species in the *Porphyrobacter* genus for which the name *Porphyrobacter mercurialis* sp. nov. is proposed; the type strain is Coronado[T] (=DSMZ 29971, =LMG 28700).

## INTRODUCTION

In this study, strain Coronado[T] was isolated from a stadium seat at Niedermeyer Field, Coronado High School in Coronado, California, USA as part of a nationwide Citizen Science project (Project MERCCURI— www.spacemicrobes.org). One goal of Project MERCCURI was to collect bacterial isolates to be used for an experiment aboard the International Space Station (ISS). The 16S rRNA gene sequenced from this particular isolate was at least 99% identical to rRNA genes from a few uncultured organisms. Uncultured isolates with high identity (=>99%) were found in samples from deep ocean sediment and the human skin microbiome. However, the highest identity to a cultured organism (*Porphyrobacter donghaensis*) (*Yoon, Lee & Oh, 2004*) was only 95.5% (as determined by BLAST, *Altschul et al., 1990*). Given the low identity to characterized species, a more detailed study of this isolate was undertaken.

Corresponding authors
David A. Coil,
coil.david@gmail.com
Jonathan A. Eisen,
jaeisen@ucdavis.edu

A phylogenetic analysis of the *Alphaproteobacteria* class led in 2005 to the creation of a new family, *Erythrobacteraceae*, to house the genera *Erythrobacter*, *Porphyrobacter* and *Erythromicrobium* (*Lee et al., 2005*). These genera were later joined by *Altererythrobacter* (*Kwon et al., 2007*) and *Croceicoccus* (*Xu et al., 2009*), the latter work also emended the description of the family. Members of the *Erythrobacteraceae* family are Gram-negative, aerobic bacteria that contain carotenoids, usually appearing pink, orange or yellow. They do not form spores, are chemo-organotrophic, and are most often associated with aquatic environments. The *Porphyrobacter* genus was established in 1993 with the description of *Porphyrobacter neustonensis*, isolated from freshwater (*Fuerst et al., 1993*). All subsequent *Porphyrobacter* species have also been isolated from aquatic sources (hot springs, seawater, and swimming pools) including *P. tepidarius* (*Hanada et al., 1997*), *P. sanguineus* (*Hiraishi et al., 2002*), *P. cryptus* (*Rainey et al., 2003*), *P. donghaensis* (*Yoon, Lee & Oh, 2004*), *P. dokdonensis* (*Yoon et al., 2006*) and *P. colymbi* (*Furuhata et al., 2013*).

Phylogenetic and biochemical characteristics presented here show that our isolate is clearly distinct from other members within the *Erythrobacteraceae* family and is most closely related to the genus *Porphyrobacter*. However, a major taxonomic revision of this family is most likely required, as has been suggested by others (e.g., *Rainey et al., 2003*; *Huang et al., 2015*). Here we describe the genotypic, morphologic, and biochemical characteristics of strain Coronado[T], based on which we propose the name of *Porphyrobacter mercurialis* sp. nov.

## METHODS

Cells were initially grown on plates containing either Reasoner's 2A agar (R2A), or lysogeny broth agar (LB). LB was made with 10 g tryptone, 10 g NaCl, and 5 g yeast extract per liter. A clear preference for growth on LB was observed and so was used for all subsequent experiments. Salt tolerance was measured by growth in liquid media (25 °C) from 0% to 20% w/v NaCl. pH tolerance was measured by growth in liquid media (25 °C) from pH 3.4 to pH 8.0. Temperature preference was measured by growth in liquid culture across the range 4 °C–30 °C. Growth in microgravity (OD600) was measured aboard the International Space Station (ISS).

Cell morphology, motility, and presence/absence of flagella were examined by light microscopy (Zeiss Axio Lab.A1) and transmission electron microscopy (TEM). Cell cultures in either exponential or stationary phase were prepared for TEM by the UC Davis Electron Microscopy lab as follows. 400 mesh copper grids with formvar/carbon support film (Ted Pella, Inc., Redding, CA) were placed on dental wax. A 10 µl drop of fixed or unfixed sample was placed onto the grid and left in a dust-free environment for 10 min. Then excess was wicked off with filter paper. A 10 µl drop of 1% PTA pH 5.8 (phosphotungstic acid) or 1% ammonium molybdate in double-distilled water was added to the grid and wicked off immediately. Grids were allowed to air-dry completely before viewing in a Philips CM120 (FEI/Philips Inc., Hillsborough, Or.) electron microscope at 80 KV.

Oxidase activity was measured using a solution of tetramethyl-p-phenylenediamine and catalase activity was measured by the addition of hydrogen peroxide to plated cells.

The hydrolysis of starch and casein were measured by standard plate methods (beef agar with soluble starch and iodine staining, and milk agar with a pancreatic digest of casein respectively). Carbon source oxidation was assayed using the Phenotypic MicroArray (TM) services offered by Biolog, Inc. using their standard procedures for gram-negative bacteria as follows. Colonies were grown on blood agar at room temperature and suspended in IF-0a inoculating fluid (Biolog) to a density of 42% transmittance. The cell suspension was diluted 1:6 in IF-0a plus 1x Dye H (Biolog) and a carbon source utilization MicroPlate (PM1; Biolog) was inoculated with 100 µl per well. The PM1 microplate was incubated at 23 °C and read by the OmniLog instrument every 15 min for 96 h. Duplicate sets of OmniLog data were converted to average read value and a threshold of 78 was required in both replicates for a positive call.

## Respiratory quinones, polar lips, and fatty acids

Cells were grown to post exponential phase (∼72 h) in 1 L of LB at 23 °C for large-scale biomass production, then centrifuged to pellet cells and lyopholized (VirTis Freeze-mobile). Analysis of respiratory quinones/polar lipids and fatty acids were carried out by the Identification Service, DSMZ, Braunschweig. Germany. Protocol details and references can be found at the following DSMZ pages; https://www.dsmz.de/services/services-microorganisms/identification/analysis-of-polar-lipids.html, https://www.dsmz.de/services/services-microorganisms/identification/analysis-of-respiratory-quinones.html, https://www.dsmz.de/services/services-microorganisms/identification/analysis-of-cellular-fatty-acids.html.

## 16S rDNA, genome sequencing, and phylogenetic analysis

Genomic DNA was extracted using a Wizard Genomic DNA Purification Kit (Promega). A nearly full-length 16S rRNA gene sequence was amplified using the 27F (5′-AGAGTTTGATCMTGGCTCAG-3′) and 1391R (5′-GACGGGCGGTGTGTRCA-3′) "universal" primers. Sanger sequencing was provided by the College of Biological Science UC-DNA Sequencing Facility (UC Davis). This DNA was also used for Illumina sequencing of the draft genome as described elsewhere (*Coil & Eisen, in press*). The genome sequence was annotated using the RAST server (*Aziz et al., 2008*; *Overbeek et al., 2013*).

The 1482bp 16S rDNA sequence was obtained from the genome assembly in RAST (GenBank: KP122961), and uploaded to the Ribosomal Database Project (RDP) (*Cole et al., 2013*). Since the RDP database is incomplete with respect to the *Erythrobacteraceae* family, additional type strain sequences were obtained from NCBI to ensure that every member of the family with official standing in nomenclature (http://www.bacterio.net/) was present in the alignment downloaded from RDP. Because the taxon names exported with this alignment contained special characters that were not compatible with phylogenetic reconstruction software, a custom script was used to remove or replace those characters with underscores. A description of and link to this script can be found in *Dunitz et al. (2015)*. The alignment was manually examined using MView (http://www.ebi.ac.uk/Tools/msa/mview/t), the secondary structure was generated using the RNAfold Web Server (http://rna.tbi.univie.ac.at/cgi-bin/RNAfold.cgi) and visualized with Forna

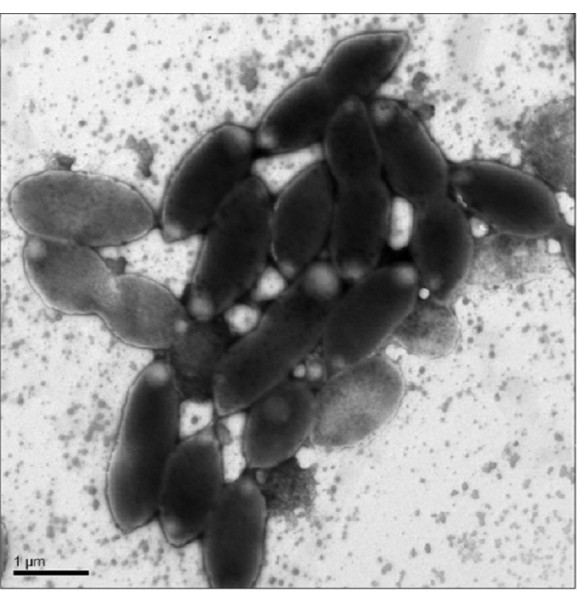

**Figure 1 Transmission Electron Microscopy (TEM) of exponential phase culture of Coronado[T] in lysogeny broth (LB), grown at 23 °C.** Cells were negatively stained with Ammonium Molybdate.

(http://nibiru.tbi.univie.ac.at/forna/). This alignment was used to generate phylogenetic trees using a variety of methods including maximum likelihood (RAxML, *Stamatakis, 2014*), Bayesian (MrBayes, *Ronquist & Huelsenbeck, 2003*; *Huelsenbeck & Ronquist, 2001*), neighbor-joining (MEGA6, *Tamura et al., 2013*, NINJA, http://nimbletwist.com/software/ninja/), and maximum parsimony (MEGA6). Dendroscope 3 (*Huson & Scornavacca, 2012*) and FigTree (http://tree.bio.ed.ac.uk/software/figtree/) were used to view and edit the phylogenetic trees.

## RESULTS AND DISCUSSION

### Morphological, physiological, and biochemical characteristics

Cells were non-motile and not observed to form spores or possess flagella, though over 30 flagella or flagella-associated genes are present in the genome. In contrast, the genome for a non-motile close relative (*Porphyrobacter cryptus*, GenBank: ASM42298v1) does not contain any flagellar genes so it is possible that strain Coronado[T] is motile under specific conditions. Unlike most members of the *Erythrobacteraceae family*, strain Coronado[T] is oxidase-negative. This strain is catalase-positive, and unable to hydrolze casein or starch.

Cells were oval or rod shaped and ranged in length from 1.2 μm to 2.2 μm with an average of 1.6 μm (Fig. 1). Cell width ranged from 0.6 μm to 1.0 μm with an average of 0.8 μm.

Growth was only observed under aerobic conditions, from 4 °C to 28 °C, with optimal growth around 25 °C. No growth was observed under microaerophilic conditions (culture caps closed). Low levels of growth were observed at pH 6.0 up to pH 8.0, maximum growth occurred around neutral pH. NaCl was required for growth, and the strain could not grow at >1.5% NaCl, optimal growth was at 0.5% NaCl. No statistically significant difference

in growth was observed between earth and microgravity aboard the International Space Station (ISS).

Strain Coronado[T] could oxidize the following as sole carbon sources: Glycyl-L-Glutamic Acid, L-Rhamnose, D-Mannose, D-Trehalose, a-D-Glucose, L-Fucose, D-Galactose, Citric acid, D-Glucuronic acid, D-Galactonic acid, L-Galactonic acid-$\gamma$-Lactone, Acetoacetic acid, Acetic acid, Pyruvic acid, and L-Malic acid.

The strain was unable to grow on N-Acetyl-D-Glucosamine, D-Saccharic Acid, Succinic Acid, L-Aspartic Acid, L-Proline, D-Alanine, Dulcitol, D-Serine, D-Sorbitol, Glycerol, D-Gluconic Acid, D,L-$\alpha$-Glycerol-Phosphate, L-Lactic Acid, Formic Acid, D-Mannitol, L-Glutamic Acid, D-Glucose-6-Phosphate, D-Galactonic Acid-$\gamma$-Lactone, D,L-Malic Acid, Tween 20, D-Fructose, Maltose, D-Melibiose, Thymidine, L-Asparagine, D-Aspartic Acid, D-Glucosaminic Acid, 1,2-Propanediol, Tween 40, $\alpha$-Keto-Glutaric Acid, $\alpha$-Keto-Butyric Acid, $\alpha$-Methyl-D-Galactoside, $\alpha$-D-Lactose, Lactulose, Sucrose, Uridine, L-Glutamine, m-Tartaric Acid, D-Glucose-1-Phosphate, D-Fructose-6-Phosphate, Tween 80, $\alpha$-Hydroxy Glutaric Acid-$\gamma$-Lactone, $\alpha$-Hydroxy Butyric Acid, $\beta$-Methyl-D-Glucoside, Adonitol, Maltotriose, 2-Deoxy Adenosine, Adenosine, Glycyl-L-Aspartic Acid, m-Inositol, D-Threonine, Fumaric Acid, Bromo Succinic Acid, Propionic Acid, Mucic Acid, Glycolic Acid, Glyoxylic Acid, D-Cellobiose, Inosine, Tricarballylic Acid, L-Serine, L-Threonine, L-Alanine, L-Alanyl-Glycine, Acetoacetic Acid, N-Acetyl-$\beta$-D-Mannosamine, Mono Methyl Succinate, Methyl Pyruvate, D-Malic Acid, Glycyl-L-Proline, p-Hydroxy Phenyl Acetic Acid, m-Hydroxy Phenyl Acetic Acid, Tyramine, D-Psicose, Glucuronamide, Phenylethyl-amine, or 2-Aminoethane.

## Phylogeny and genome analysis

Phylogenetic analysis was performed using the full length (1,482 bp) 16S rDNA sequence from the genome assembly, not the shorter (1,350 bp) version from Sanger sequencing. The Coronado[T] 16S rDNA sequence showed less than 95.5% identity to other *Porphyrobacter* species and identity is even lower to other genera in the family. Given the low 16S rDNA identity to other members of the family, we did not perform DNA–DNA hybridization as this would have been uninformative (*Stackebrandt & Goebel, 1994*; *Tindall et al., 2010*; *Meier-Kolthoff et al., 2013*).

Phylogenetic trees built by varying the alignment and tree-building algorithms, number of taxa included, and choice of outgroup demonstrated both that the current taxonomy of the family is in need of revision (as has been suggested by others, e.g., *Rainey et al., 2003*; *Huang et al., 2015*) and that the placement of *Coronado*[T] within the family is not stable. Because the Bayesian and Maximum Likelihood methods show a very similar topology, and are considered the most accurate methods for phylogenetic analysis (e.g., *Yang & Rannala, 2012*; *Kuhner & Felsenstein, 1994*; *Hall, 2005*), we have shown those trees in Figs. 2 and 3. Both of these trees place Coronado[T] within the *Porphyrobacter* clade, though in the Bayesian tree there is a polytomy at the base of this clade. We note, as also shown recently by *Huang et al. (2015)*, that this clade is always polyphyletic with respect to *Erythromicrobium ramosum* and often to *Erythrobacter litoralis*. In addition, Coronado[T] is

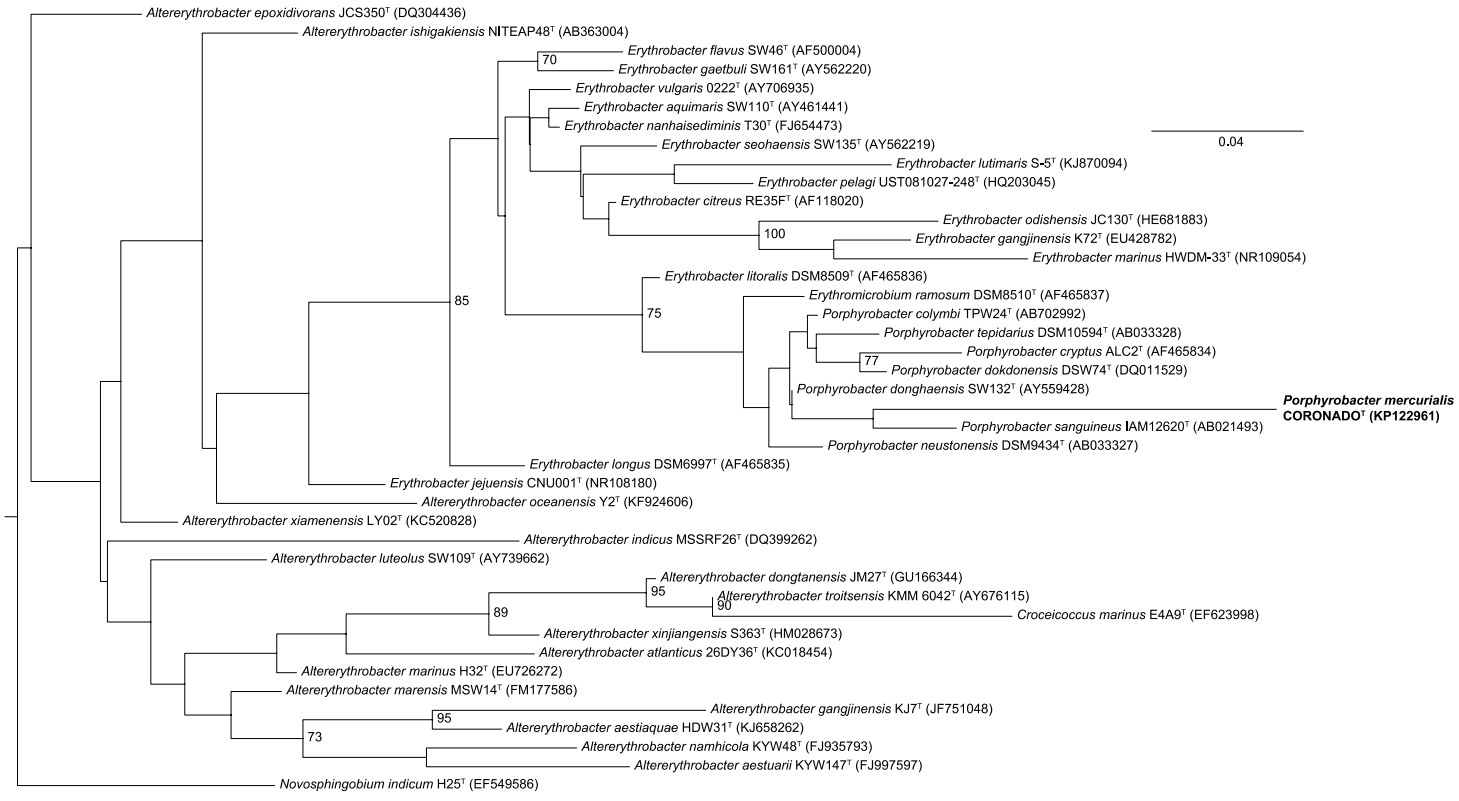

**Figure 2 Maximum Likelihood tree based on the 16S rRNA gene sequence of Coronado^T and all type strains from the *Erythrobacteraceae* family.** The tree was inferred from an Infernal-based alignment created in RDP using RAxML with the gamma model of rate heterogeneity. Numbers at the nodes (only values >70 are shown) represent support values for 1,000 bootstrap replicates. The tree was rooted to *Novosphingobium indicum* as an outgroup since this species was shown to be one of the closest relatives to the *Erythrobacteraceae* family in a tree of all *Alphaproteobacteria*.

found on a long branch due to several changes that are unique to this strain, relative to the rest of the family. These changes are identical in both the assembly and the Sanger sequence and are all compatible with the secondary structure model of 16S (e.g., changes in a stem nucleotide pair with the appropriate base). Based on this analysis, we chose to compare Coronado^T to the five other *Porphyrobacter* species listed in Table 1.

Analysis of the draft genome of strain Coronado^T was used to complement the physical characterizations typical of the family *Erythrobacteraceae* and the genus *Porphyrobacter*. For example, Coronado^T does not contain any of the numerous genes involved in bacteriochlorophyll biosynthesis, rendering protein extraction/spectrophotometry unnecessary. Conversely, while no flagella were observed by TEM, this strain appears to possess the required genes making it likely that the flagella were lost in sample preparation or that their expression is condition-dependent.

## Polar lipid, respitory lipoquinone, and fatty acid methyl esters

The major cellular fatty acids of strain Coronado^T are C18:1$\omega$7cis (56.6%) and C16:0 (20.3%). Other fatty acids found in significant amounts (>1%) are 2-OH-C14:0 (4.8%), C16:1$\omega$5cis (1.1%), C16:1$\omega$7cis (9.8%), C17:1$\omega$6cis (2%), C18:1$\omega$5cis (1.1%), and

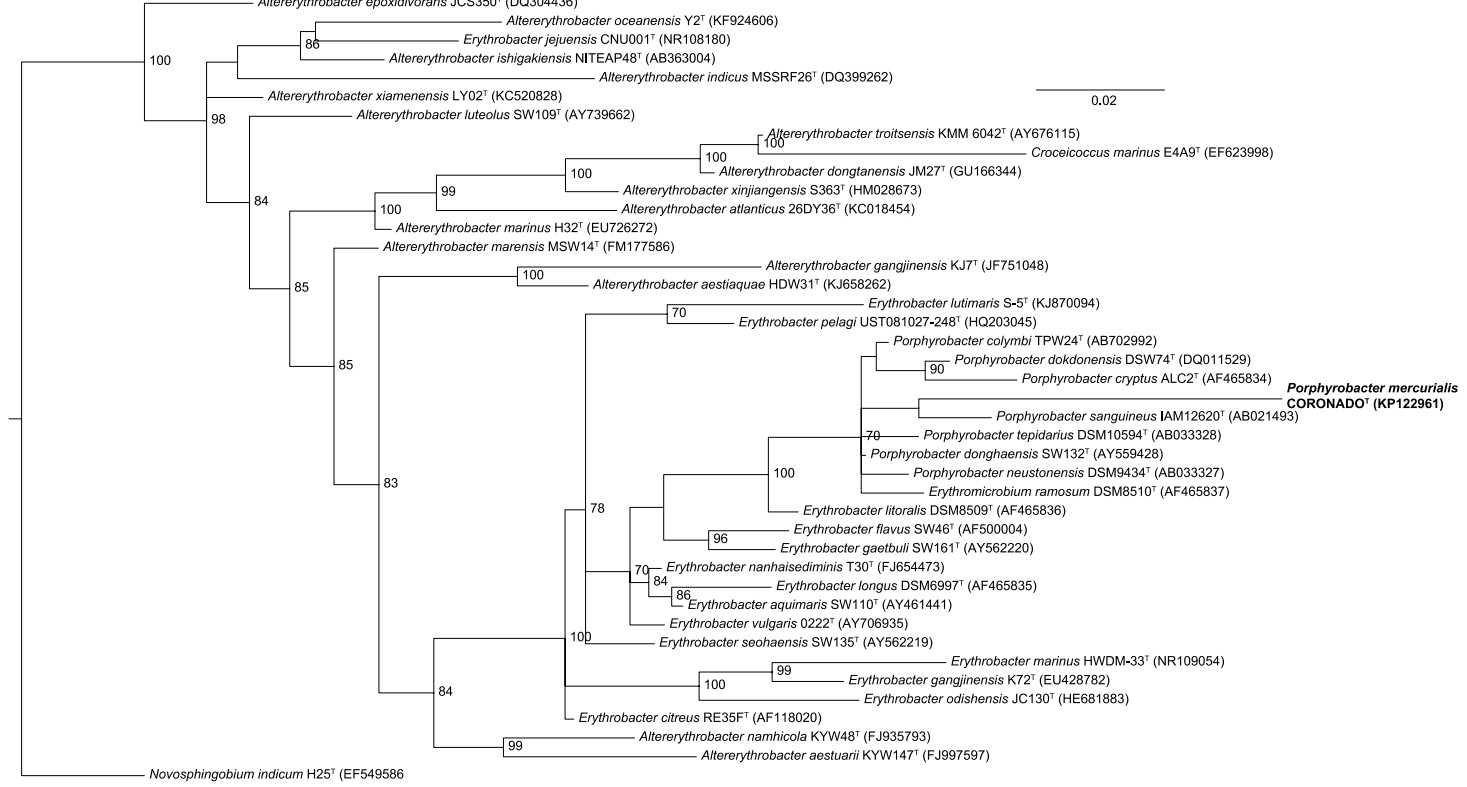

**Figure 3** Bayesian tree based on the same sequence alignment described in **Fig. 2** (**Infernal-based RDP alignment of 16S rRNA gene sequence of CoronadoᵀT and all type strains from the *Erythrobacteraceae* family**). Tree was inferred using MrBayes with the General Time Reversible (GTR) substitution model. Numbers at nodes (only values >70 are shown) represent posterior probability values from 1,000,000 iterations of the tree, with a 25% relative burnin. The tree was rooted to *Novosphingobium indicum* as an outgroup since this species was shown to be one of the closest relatives to the *Erythrobacteraceae* family in a tree of all *Alphaproteobacteria*.

C18:0 (1.2%). The fatty acid profile of strain CoronadoᵀT fits generally within the ranges described for members of the most closely related genera (*Erythrobacter*, *Porphyrobacter* and *Erythromicrobium*, comparison data from *Hiraishi et al., 2002*). The two exceptions to this are a slightly higher level of C14:0 than average and a much higher level of C16:0 than average. However, it is difficult to compare across studies since variation in growth conditions can significantly influence the fatty acid profile.

The major respiratory quinone is ubiquinone 10 (92%), as it is for all members of the *Erythrobacteraceae* family. The predominant polar lipid is phosphatidylglycerol, with significant amounts of sphingoglycolipid and phosphatidylethanolamine. Smaller amounts of diphosphatidylglycerol, phosphatidylcholine, and two unidentified phospholipids were also observed (Fig. 4).

## Conclusions

Strain CoronadoᵀT clearly falls within the *Erythrobacteraceae* family, based on both phylogenetic analysis and chemotaxonomic and molecular characteristics (notably fatty acid profile, carotenoid production, major respiratory quinone, and GC content). Using Bayesian and Maximum Likelihood phylogenetic reconstruction, the strain falls within a

**Table 1** Phenotypic comparison of Coronado[T] and other members of the *Porphyrobacter* genus.

| Characteristic | Strain Coronado[T] | P. sanguineus | P. tepidarius | P. donghaensis | P. cryptus | P. neustonensis |
|---|---|---|---|---|---|---|
| Cell shape[a] | O/R | R | O/R | C/O/R | R | C/O/R |
| Color of colony[b] | YO | OR | O | OR | OR | OR |
| Motility | – | + | – | – | + | + |
| Presence of BChl *a* | – | + | + | + | + | + |
| Growth in NaCl (%): | | | | | | |
|     Range | .1–1.5 | ND | 0.0–1.3 | ND | ND | ND |
|     Optimum | .5 | 1 | ND | 2 | ND | ND |
| Growth pH: | | | | | | |
|     Range | 6.0–8.0 | ND | ND | ND | 6.0–9.0 | ND |
|     Optimum | 7.0 | 7.0–7.5 | ND | 7.0–8.0 | 7.5–8.0 | ND |
| Growth temperature (°C) | | | | | | |
|     Range | 4–28 | 20–37 | ND | 10–45 | ND | 10–37 |
|     Optimum | 25 | 30 | 40–48 | 30–37 | 50 | ND |
| Catalase | + | + | ND | + | + | + |
| Oxidase | – | + | – | + | + | – |
| Major cellular fatty acids | $C_{18:1}\omega 7cis$, $C_{16}$ | $C_{18:1}\omega 7cis$ | ND | $C_{18:1}\omega 7cis$, $C_{17:1}\omega 6cis$ | $C_{18:1}\omega 7cis$ | n18:1 |
| DNA G + C content | 67.3 | 63.8–64 (mol%) | 65 (mol%) | 65.9–66.8 (mol%) | 66.2 (mol%) | 65.7–66.4 (mol%) |

**Notes.**
[a] C, Cocci; O, oval; R, rod.
[b] Y, Yellow; O, orange; R, red.
  Positive, +; negative, −; No data available, ND. Data from this study and *Hiraishi et al. (2002)*, *Rainey et al. (2003)*, *Hanada et al. (1997)*, *Yoon, Lee & Oh (2004)*, *Fuerst et al. (1993)*, *Yoon et al. (2006)*.

well-supported (but polyphyletic) *Porphyrobacter* clade. In addition, Coronado[T] shares a number of characteristics with *Porphyrobacter*, including the fatty acid profile, polar lipid composition, catalase activity, etc. The largest differences are the tolerance for growth at lower temperatures, elevated C16:0, and the lack of bacteriochlorophyll *a* (for the latter of which we have proposed emending the genus description). These characteristics, in combination with the phylogenetic analysis, lead us to propose that Coronado[T] be classified as *Porphyrobacter mercurialis* sp. nov. In the future, the taxonomic status of this strain may change depending upon availability of clear and distinctive evidence for a new genus as per polyphasic taxonomic and/or genome sequence based taxonomic approaches.

### Emended description of the genus Porphyrobacter *Fuerst et al., 1993*

The description is identical to that given by *Fuerst et al. (1993)* with the following amendments. Most species synthesize BChl *a* on low-nutrient media under aerobic and semiaerobic conditions. DNA base composition is 63.8–67.3 G+ C.

### Description of Porphyrobacter mercurialis sp. nov

*Porphyrobacter mercurialis* (mer.cur.ia.al'is L. adj. *mercurialis*, temperamental in reference to difficulties in establishing consistent growth requirements during characterization).

    Gram-negative, non-spore forming pleomorphic bacteria. Strictly aerobic and chemoheterotrophic. Contains carotenoids, but not bacteriochlorophyll *a*. The major

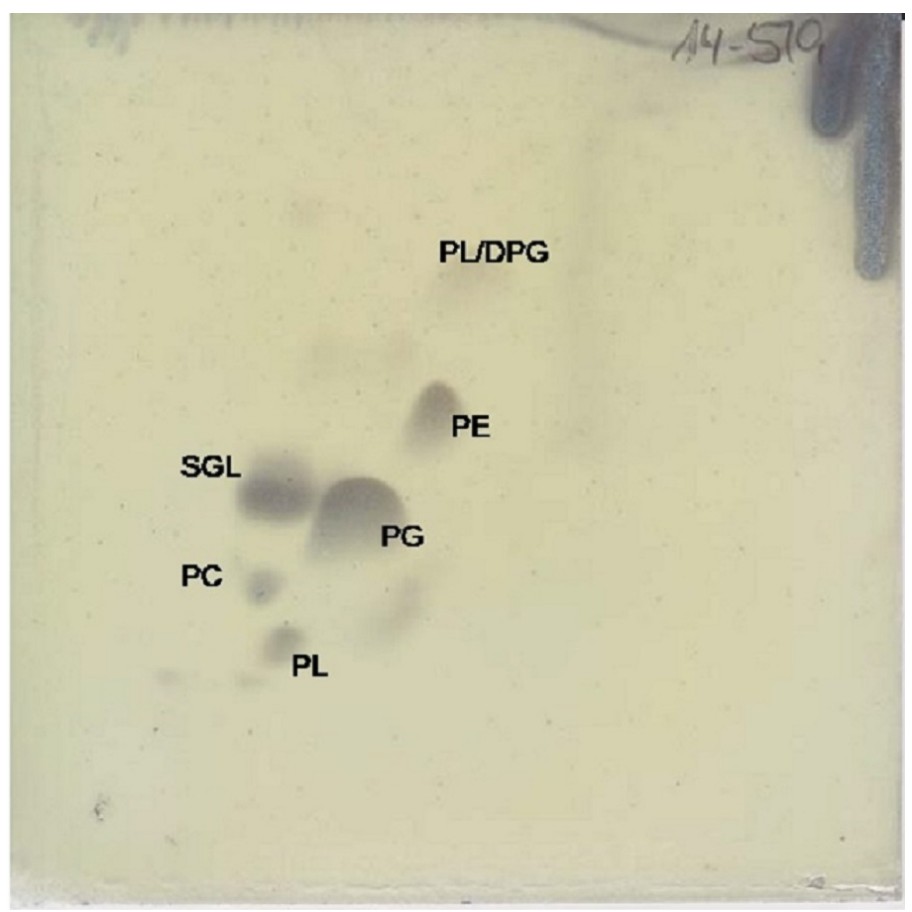

**Figure 4  Two dimensional silica gel thin layer chromatography results for polar lipids, isolated following standard protocols at the DSMZ (see 'Methods').** PL, phospholipid; PG, phosphatidylglycerol; PC, phosphatidylcholine; PE, phosphatidylethanolamine; DPG, diphosphatidylglycerol; SGL, Sphingo-glycolipid.

respiratory quinone is ubiquinone 10, the dominant phospholipds are phosphatidyl-glycerol, sphingoglycolipid, and phophatidylthanolamine. Colonies on LB are round, glossy, and less than 1 mm in diameter. Coloration on agar and in dense liquid culture is dark yellowish orange, appears lighter yellow in liquid culture during exponential phase. Morphology is non-motile pleomorphic ovals or short rods (average 1.6 μm length), often with pointed ends. Growth in liquid culture from 4 to 28 °C, optimum growth at 25 °C. Requires salt for growth, optimal growth at 0.5% NaCl, cannot grow at 3% NaCl. Tolerates up to pH 8.0, optimum around neutral pH. Oxidase-negative and catalase-positive, does not hydrolyze casein or starch. Does not grow on R2A agar. The primary fatty acids are C18:1ω7cis and C16:0, with a high percentage of the latter relative to other species in the *Erythrobacteraceae* family. Can oxidize the following as sole carbon sources: Glycyl-L-Glutamic Acid, L-Rhamnose, D-Mannose, D-Trehalose, a-D-Glucose, L-Fucose, D-Galactose, Citric acid, D-Glucuronic acid, D-Galactonic acid, L-Galactonic acid-$\gamma$-Lactone, Acetoacetic acid, Acetic acid, Pyruvic acid, and L-Malic acid.

The type strain Coronado$^T$ (=DSMZ 29971, =LMG 28700) was isolated from a stadium seat in Coronado, California, USA. The GC content of the type strain is 67.3%, as determined by genome sequencing. The genome size is approximately 3.5 MB.

## ACKNOWLEDGEMENTS

The authors would like to thank the Coronado Pop Warner Islanders for initial collection of the sample and participation in Project MERCCURI, as well as Kris Tracy who assisted in the etymology of the proposed species name.

### Funding

This work was funded by a grant to JAE from the Alfred P. Sloan Foundation. The funders had no role in study design, data collection and analysis, decision to publish, or preparation of the manuscript.

### Grant Disclosures

The following grant information was disclosed by the authors:
Alfred P. Sloan Foundation.

### Competing Interests

Jonathan Eisen is an Academic Editor for PeerJ.

### Author Contributions

- David A. Coil conceived and designed the experiments, performed the experiments, analyzed the data, wrote the paper, prepared figures and/or tables, reviewed drafts of the paper.
- Jennifer C. Flanagan conceived and designed the experiments, performed the experiments, analyzed the data, reviewed drafts of the paper.
- Andrew Stump performed the experiments, prepared figures and/or tables, reviewed drafts of the paper.
- Alexandra Alexiev performed the experiments, reviewed drafts of the paper.
- Jenna M. Lang analyzed the data, wrote the paper, prepared figures and/or tables, reviewed drafts of the paper.
- Jonathan A. Eisen reviewed drafts of the paper.

### DNA Deposition

The following information was supplied regarding the deposition of DNA sequences:
GenBank: KP122961.1.

### Data Availability

The research in this article did not generate any raw data.

## New Species Registration

The following information was supplied regarding the registration of a newly described species:

Germany (DSMZ 29971)

Belgium (LMG 28700)

Certificates of deposit available if needed.

## Supplemental Information

Supplemental information for this article can be found online at http://dx.doi.org/10.7717/peerj.1400#supplemental-information.

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
