# Peer review of "Porphyrobacter mercurialis sp. nov., isolated from a stadium seat and emended description of the genus Porphyrobacter"

_PeerJ, doi:10.7717/peerj.1400_

## Round 0.1 · original submission · Major Revisions

Reviewers suggest that the proposal of novel genus for strain Coronado is not acceptable. However, proposal of a novel species of Porphyrobacter with emendation of the description of genus Porphyrobacter may be considered for publication. Please revise your manuscript according to the referees’ comments or provide rebuttals.

Reviewer 1 ·

Basic reporting

I have not identified any area where the article fails to meet PeerJ standards.

Experimental design

It is ok!

Validity of the findings

Proposal of a new genus is not justified by the data presented but assignment to an established species may be ok

Additional comments

General comments:
In a taxonomic study where a novel species (and genus) is proposed data (mainly phenotypic) have to be collected in order to characterize the species. Based on important differences in phenotypic traits (mainly chemotaxonomic characteristics which are much more conserved than physiological and biochemical traits which are more useful for species differentiation) the proposal of a novel species/genus may be concluded. In this submission I do not see a single argument which is justifying the proposal of a novel genus. I do admit that in the case of Coronado it is a sophisticated task to find a genus for this strain because phylogenetically species of different genera are intermingled with each other. Proposal of a novel genus appears to be a clever solution of the problem but, since no phenotypic data are supporting the novel genus proposal of a novel genus is not acceptable because it will cause more confusion within this family. From my point of view the novel species could be assigned to the genus Porphyrobacter, although it lacks genes related to biosynthesis of bacteriochlorophyll. The species is phylogenetically placed at the root of a clade comprising mainly species of the genus Porphyrobacter, including Porphyrobacter neustonensis the type species of the genus. Furthermore, among type species of genera of the family Erythrobacteraceae, P. neustonensis shares highest 16S rRNA gene sequence similarity with strain Coronado. The disagreement with the genus description, not producing bacteriochlorophyll, can be overcome by emending the genus description of Porphyrobacter, saying that BCHL a is produced by the majority of species but some species may lack this ability. That this can be done was shown In the neighboring genus Erythrobacter which originally also was described to produce BChl. However, several species were described as members of the genus Erythrobacter though they lack the ability to produce BChl a, including Erythrobacter citreus and Erythrobacter flavus.
It is unclear how the authors did come to the result that the highest 16S rRNA gene sequences similarity with a species of the family Erythrobacteraceae is 95% (P. sanguineus). When I carried out a 16S rRNA gene comparison (using the ezbiocloud server, worldwide applied predominantly by bacterial taxonomists) with the sequence of Coronado the type strains of the species Porphyrobacter colymbi, Porphyrobacter donghaensis, Alterythrobacter xinjangensis, Porphyrobacter dokdonensis, Porphyrobacter neustonesis and Porphyrobacter tepidarius all shared similarity values higher than 95% with strain Coronado (95.49, 95.43, 95.22, 95.0, 95.01, and 95.01 %, respectively). Numerous species of the family are not included in the phylogenetic study including two species sharing >95 % 16S rRNA gene sequence similarity, namely P. colymbi and Alterythrobacter xinjangensis. It is mandatory to include these two species to the phylogenetic analysis and very good argument have to be provided for not including all species of the family including the monospecific genus Croecicoccus.

At the end of the text body the authors should add some sentences why they believe that the isolate is representing a novel species and why they propose to assign it to genus X.
Specific comments.
Line 72: What is ‚DDH2O ‘. I guess that double distilled water is meant. The ‘2’ should be written as a subscript. Abbreviation should be only used when they have been explained somewhere.

Line 75-76: It is not acceptable to write that these analyses were carried out by standard methods. For instance, only for test of oxidase at least four methods are standard. Please provide details how these tests were carried out.
Line 76: As far as I know Biolog phenotypic MicroArray is not testing utilization of carbon sources but only oxidation of carbon sources.
Line 80: please replace ‘utilization’.

Lines 85-88: Some details how analyses were carried out including referencing original papers Furthermore, it is most important to indicate the physiological age of biomass that was subjected to fatty acid analysis because the resulting fatty acid profile can differ significantly based on the physiological age of biomass. Resulting data are only useful in future studies when relatives of the novel species will have to be compared to each other, when the new strain can be harvested at the same physiological age.
In taxonomic papers dealing with the proposal of novel species it is common practice to calculate phylogenetic trees on two better three algorithms (maximum likelihood, maximum parsimony, neighbor joining).
Line 113: if a species name is used in the submission for the first time, the genus name should not be abbreviated.

Line 117, 118: not ‘um’ should be used but ‘µm’ to indicate the size of cells.
Line 119: I could not find any information in ‘Methods’ how relation to oxygen was tested. Did the authors try to grow the strain anaerobically or in the presence of 5% CO2?
Line 121: What salt was required?

Line 124: ‘….utilize…..’ should be replaced by ‘… oxidize….’
Line 158: ‘chlorophyll’ should be replaced by ‘bacteriochlorophyll’
Line 169: The authors are pointing on certain difference in the fatty acid profile compared to related genera. However, taxonomic conclusion can be only drawn based on fatty acid profiles if the fatty acids were analyzed from biomasses that were harvested at the same physiological age and grown under the same conditions (medium composition, temperature, pH, NaCl concentration, etc.) which may influence the fatty acid profile. Since this is not the case here the reported differences are without any significance.

Line 186: When the etymology is provided it is not sufficient to translate the Latin name. It also has to be shortly explained why the authors consider the species of be temperamental.
Fig. 2: In the phylogenetic tree the 16S rRNA gene sequence accession number should be provided with the strain like with all other entries.

Fig. 3: In the image showing the polar lipid profile several spots are visible which have not been labelled. Even if these lipids do not contain a sugar moiety or a phosphate or amino residue they are showing the presence of lipids which should be labelled as well.
Fig. 3: What is the rationale to give one spot the label PL/DPG. This spots shows exactly the chromatographic motility and shape which has been shown in numerous publications for DPG.

Reviewer 2 ·

Basic reporting

Authors have made an attempt and structured the manuscript nicely but should have taken few more essential steps justify the description of novel genus

Experimental design

Description of a novel genus needs a vast comparative polyphasic study to differentiate the proposed type strain with all existing genera of respective family, which could not be done by the authors. Phenotypic comparison (done in the authors laboratory) with closest neighbours was missing. Authors could not justify the delineation of novel genus based on differentiating phenotypic characteristics amongst the family Erythrobacteriaceae.

Validity of the findings

Consistency of the results were missing in case of closest neighbor and incubation temp. It is noticed that authors did not provide GenBank accession number of the proposed type strain which was a major slip.

Additional comments

The manuscript needs major revision to consider for the further review and processing.
1.For novel genus description, proposed strain should have genus specific differentiating characteristics to all the existing genera of respective family, which is lacking in the MS. Authors should do comparative study with the members of the closest genera in their laboratory and to compare at least genus specific chemotaxonomic characteristics such as polar lipids, quionones, fatty acids and justify their description based on differentiating characteristics. Genomic difference such as 16S rRNA gene dissimilarity alone cannot support the genus description. Additionally, analysis of housekeeping genes also play a role in delineating novel genus. (Tindall et al., 2010, Notes on the characterization of prokaryote………..purposes; Logan eta al., 2009, Proposed minimal standards….bacteria; though the paper is for endospore forming bacteria, standards applicable to non-spore forming bacteria also).
Until unless, authors could show the genus specific differentiating characteristics, strain can be accepted as a novel genus. Otherwise they should restrict its description to the species level after the comparative study with the phylogenetically nearest species.
2. No GenBank accession number for the proposed novel genus is given in the MS. Since there is no access to the sequence online, it is impossible to review the extent of similarity with phylogenetic neighbors. It is mandatory to submit GenBank accession number for novel taxa descriptions (Tindall et al., 2010. Notes on the characterization of prokaryote………..purposes).
3. Phylogenetic trees should be drawn with three different methods, such as NJ, ML, MP to check the consistency of the clustering pattern of proposed strain and should be added as supplementary data or mention in the Manuscript.
4. It is strongly recommended to have multiple strains for the description of novel taxa.

Minor comments
(1) Line 47 & 147 Surprising to know two different closest neighbors “Porphyrobacter donghaensis” in Line 47 and “Porphyrobacter sanquineus” in Line 147; Explain?
(2) Lines
61,62,81,85,120 Any logic behind the variation in the incubation temperature?
(3) Line 76-77 Give reference
(4) Table Fatty acids… C16..???, references for strains used for comparision???
(5) References Make all the references in a uniform format and as per the journal regulations

Reviewer 3 ·

Basic reporting

The authors have tried to establish a strain designated as CoronadoT as novel genus. Although, the strain occupies a clade well separated from Porphyrobacter and Erythrobacter clusters, (and there are two Erythrobacter spp. within the cluster of different spp of Porphyrobacter). The extent of 16S rRNA gene sequence identity (95.45%) with closest phylogenetic relative (Porphyrobacter colymbi) is low but in absence of sufficient phenotypic differences, it cannot be considered as a new genus.
Comparative polar lipid data may be very useful. If it reflects sufficient differences from the same for type species of different close genera (within the family), authors may still go for a new genus, provided, they point out major phenotypic differences, following fresh comparative analysis with representative type strains of type species of close genera. For new genus, authors may also carry out phylogenetic tree construction using different methods.
The strain may however be considered as new sp. of Porphyrobacter.
But to establish that authors should carry out:
1. Comparative phenotypic analyses of CoronadoT with its closest phylogenetic relatives to establish sufficient phenotypic differences.
2. Phylogenetic tree showing position of the strain among various members of the genus Porphyrobacter.
3. Tree must reflect accession No. of type strain as well as accession no. of 16S rRNA gene for the same. This will be easier for workers who will be using tree data for future taxonomic work. This will in turn increase citation of the paper.
4. There are not sufficient phenotypic data as far as its differences with respect to close phylogenetic spp. are concern. Definitely, it cannot be considered as a novel genus.
5. Discussion part is lacking. Manuscript should have proper discussion section reflecting novelty (sp. nov. or gen. nov) of the strain.
6. The authors may please contact a specialist regarding etymology of the proposed new name of the bacterium.

Experimental design

1. Authors should have carried out comparative study of phenotypic properties of type strains (at least the closest one, Porphyrobacter colymbi and already reported data for the type species of this genus Porphyrobacter neustonensis) of related spp.
2. Polar lipid study was carried out but it has no value until and unless it is compared/ comparable to closest phylogenetic relatives. A comparative analysis with representative type strain of type species of closest genera would have given better idea as far as differences are concerned.
3. Carotenoids are also reported in the genus Porphyrobacter (P. dokdonensis), along with bacteriochlorophyll a. So, this may be a species specific character.
4. The Starch and casein hydrolysis properties for the strain CoronadoT should be compared with close phylogenetic relatives.
5. Oxidase negative is reported for Porphyrobacter neustonensis and P. tepidarius (Pl refer IJSEM 2006: 56, 1079-1083).

Validity of the findings

The finding made in the manuscript is all right. From phylogenetic point of view (16S rRNA gene sequence analyses) the strain looks a probable new genus. But as per current consensus, this has to be established following polyphasic taxonomic approach. The novelty has to be established in such a way that there are sufficient differences in properties considered as per polyphasic approach.
Comparison of well defined different phenotypic properties among close phylogenetic relatives are lacking in the manuscript. That is why it cannot be considered as new genus even a new species. For the latter author may carry out comparison with close spp.

Additional comments

6. Line 23 – 24, do the author mean cluster or clade?. This line should be rephrased. Oxidase negative is already reported in P. neustonensis and P. tepidarius (Pl refer IJSEM 2006: 56, 1079-1083).
7. The Phenotypic characters (line 25-26), looks similar to the members of the genus Porphyrobacter except few (i.e. Growth at 4C, etc.). These may be taken into consideration when going for new species description.
8. Other than C14:0, there was no unique fatty acid for CoronadoT, The difference in fatty acid profile looks quantitative, if is compared with other representative type strains of different spp. of the genus Porphyrobacter.
9. Table 1. Erythrobacter and not Erthrobacter.

---

## Round 0.2 · Minor Revisions

Now the manuscript is suitable for publication. There are a few minor revisions requested by the reviewers. In CoronadoT, T should be superscript throughout the manuscript. Similar formatting should be applied with accession number.

Reviewer 2 ·

Basic reporting

NA

Experimental design

NA

Validity of the findings

NA

Additional comments

Authors have improved and made changes as per the previous suggestions given, to the manuscript. Few typographical corrections to be made by the authors.

Line No. Comments
39 Correct cartenoids, to “ Carotenoids ”
87 I am doubt that RAST server is only for protein annotation of whole genome & not for rRNAs, if so, what other servers used for rRNA predictions?
References
221 & 266 Correct “nov” to Nov and ‘oct’ to Oct, to maintain uniformity
Italicize the bacteria names in the references given

Reviewer 3 ·

Basic reporting

The manuscript now looks well connected and a clear conclusion was drawn with respect to phenotype. The authors also have also fulfilled various criteria needed for a novel sp. description. So, in my opinion, the manuscript can now be considered for publication.
There are some minor points that may be corrected to make the manuscript better readable.

Experimental design

The experiments conducted and discussed now meets the basic standards for description of a novel species in Prokaryotic domain of life.

Validity of the findings

The findings reported in the manuscript are authentic and meets the standard of this journal and for authentic description of species under the domain Bacteria.

Additional comments

The manuscript now looks well connected and a clear conclusion was drawn with respect to phenotype. The authors also have also fulfilled various criteria needed for a novel sp. description. So, in my opinion, the manuscript can now be considered for publication.
There are some minor points that may be corrected to make the manuscript better readable.
1. Authors are requested to give an outline of different spp. of Porphyrobacter currently known to scientific world with proper references, in the introduction section. This is lacking.
2. Sentences between line 29 to 35 “A cotton swab was used …………………………. Isolate was undertaken” looks out of place. It can either be shifted to last Para before the sentence “Here we report------------sp. nov.“ in the introduction section or can be rephrased and shifted to results & discussion part.
3. Please, rewrite the sentence “A 2005 phylogenetic analysis of the Alphaproteobacteria class led to the creation of a new family, Erythrobacteraceae, to house the genera Erythrobacter, Porphyrobacter and Erythromicrobium [15].” As follows:
Phylogenetic analysis of Alphaproteobacteria class (carried out during 2005) led to creation of a new family Erythrobacteraceae, to house the genera Erythrobacter, Porphyrobacter and Erythromicrobium [15].
4. Please write the scientific name in italic form, wherever necessary (for example, refer to Table 1).
5. Line 169-170: Please replace ‘phylogenetic analysis and physical characteristics’ by phylogenetic analysis, chemotaxonomic and molecular characteristics.
6. Authors may replace the last sentence Line 176- 178 “We are aware that a future in depth polyphasic………………………… this strain” by Although, phylogenetic analyses clearly indicated the strain CoronadoT to be a possible new genus, however, in absence of distinct phenotypic and chemotaxonomic evidences, the strain was concluded as a novel sp. In future, the taxonomic status of this strain may change depending upon availability of clear and distinctive evidences as per polyphasic taxonomic and/or genome sequence based taxonomic approaches.
7. Please consult any standard paper from IJSEM journal to give final finishing, as far as descriptive information is concerned.
8. Please confirm etymology of the new specific epithet with some expert.
9. Please add the NJ tree as supplementary figure.

---

## Round 0.3 · accepted · Accept

The manuscript is suitable for publication.